



# Variation in bacterial composition, diversity, and activity across different subglacial basal ice types

Shawn M. Doyle[1], Brent C. Christner[2]

[1]Department of Oceanography, Texas A&M University, College Station, TX, 77843, USA

[2]Department of Microbiology and Cell Science, University of Florida, Gainesville, FL, 32611, USA

*Correspondence to*: Shawn M. Doyle (shawndoyle@tamu.edu)

**Abstract.** Glaciers and ice sheets possess layers of basal ice characterized by high amounts of entrained debris that can serve as sources of nutrients and organic matter, providing habitat for microorganisms adapted to the frozen conditions. Basal ice forms through various mechanisms and is classified based on its ice and debris content; however, little is known about

variation in microbial composition, diversity, and activity across different basal ice types. We investigated these parameters in four different types of basal ice from a cold-based and temperate glacier and used a meta-analysis to compare our findings with microbiome studies from other frozen environments. We found basal ice environments harbor a diverse range of microbiomes whose composition and activity can vary significantly between basal ice types, even within adjacent facies from the same glacier. In some debris-rich basal ices, elevated ATP concentrations, isotopic gas signatures, and high ratios

of amplified sequences for 16S rRNA relative to that for 16S rRNA genes implicated certain bacterial taxa (e.g., *Paenisporosarcina*, *Desulfocapsa*, *Syntrophus*, *Desulfosporosinus*) as being potentially active, with ice temperature appearing to be an important predictor for the diversity of taxa inferred to be active. Compared to those of other sympagic environments, these microbiomes often resembled those found in permafrost or perennial cave ice rather than other glacial ice environments. In contrast, debris-poor basal ices harbored microbiomes more like those found in oligotrophic englacial

ice. Collectively, these results suggest that different basal ice types contain distinct microbiomes that are actively structured by the diagenesis of their habitat.

## 1. Introduction

At the ice-bed interface of a glacier or ice sheet, basal ice forms when subglacial debris and/or water is incorporated into the base of the ice mass through mechanisms such as freeze-on, regulation, or incorporation of pre-existing ice (Knight, 1997;

Souchez et al., 2006). Basal ice has a physical structure, chemistry, and microbial composition that is directly affected by its proximity to and interaction with material beneath the glacier (Knight, 1997; Doyle et al., 2013; Montross et al., 2014). When frictional or geothermal heating create temperate conditions at the bed, the melting of basal ice can provide inputs of organic matter, nutrients, oxidants, and microorganisms to the subglacial environment (Siegert et al., 2001). Hence, basal ice likely represents an important vector for dispersing microbes in subglacial aquatic environments that exist within the

glacier's flow path (Achberger et al., 2017).





Basal ice layers contain debris entrained from the underlying substrate that ranges in size from fine-grained clays and silts to coarse sands, gravels, and boulders. Basal ice formation can produce layering and stratification patterns that become deformed by the folding and shearing forces associated with glacier flow (Samyn et al., 2008). These processes of entraining
and restructuring of subglacial debris combine to produce distinct types of basal ice. This ranges from relatively debris -poor types containing <1% (w/v) sediment to debris-rich types composed almost entirely of sediment with only interstitial ice (Knight, 1997). When compared to the overlying englacial ice (i.e., derived from snowfall), the sediment-rich basal ice contains higher concentrations of potential microbial substrates that include formate, acetate, ferrous iron, and ammonia (Skidmore et al., 2000; Wadham et al., 2004; Tung et al., 2006; Yde et al., 2010). These compounds are hypothesized to be
an important source of electron donors for heterotrophic and lithotrophic microorganisms inhabiting subglacial environments, including frozen matrices within the basal ice itself (Montross et al., 2014, 2013).

Although ecosystems associated with sympagic environments were first described nearly 60 years ago (Bunt, 1963), viable microorganisms entrapped within glacial ice have been assumed to exist in a persistent state of metabolic dormancy (Abyzov
et al., 1998). This view has been challenged based on evidence from several pioneering studies that revealed anomalies in entrapped gas concentrations within englacial (Sowers, 2001; Campen et al., 2003) and basal ice (Souchez et al., 1995, 1998; Price and Sowers, 2004). The metabolic activity of microorganisms in basal ice is now thought to play an important role in the biogeochemistry of downstream terrestrial, freshwater, and marine environments that receive seasonal discharges of basal ice and subglacial meltwater (Wadham et al., 2010; Barker et al., 2018; Hopwood et al., 2019; Vick-Majors et al.,
2020). In fact, transport of basal material is one of the largest sources of labile dissolved organic carbon and iron to marine systems receiving outflow from glaciated watersheds (Hood et al., 2009; Bhatia et al., 2013; Hawkings et al., 2020).

The world's ice sheets are estimated to contain ~10.2 Pg of organic carbon (Priscu et al., 2008), which represents only ~0.5% of the global soil organic carbon pool (Batjes, 2014). However, the sediments underlying the Greenland and Antarctic ice
sheets may contain reservoirs of ~500 and ~21,000 Pg of organic carbon, respectively (Wadham et al., 2008, 2012). Due to accelerated rates of melt and glacial retreat in the polar regions, the products of organic matter processing by subglacial microorganisms may play an increasingly important role in global carbon cycling. For example, the subglacial production of biogenic methane and its subsequent release to the atmosphere could represent a positive feedback that accelerates further climate warming (Wadham et al., 2008). Indeed, basal ice sediments have been shown to contain viable methanogens
(Skidmore et al., 2000; Boyd et al., 2010) as well as methanotrophs (Dieser et al., 2014; Michaud et al., 2017), indicating the presence of a subglacial methane cycling. Microbial activities in the basal zones of the world's glaciers and ice sheets may thus be an overlooked component of Earth's climate oscillations over geologic timeframes. Despite their potential importance, the logistic challenges associated with accessing these environments have contributed to the dearth of information on the microbes and biogeochemical processes they perform in the basal zone of large ice masses.


In this study, we used 16S rRNA gene amplicon sequencing to profile the composition and structure of microbial assemblages within basal ice types and from different locations. To explore how these microbial assemblages compare to those in other perennially frozen environments, we performed a meta-analysis that combined our data with that from other studies. Many of these studies sequenced different variable regions of the 16S rRNA gene (e.g., V4, V2, V3V4) that do not

fully overlap, hindering the use of traditional OTU-based approaches. To address this, we used the SATé-enabled phylogenetic placement (SEPP) technique to insert exact amplicon sequence variants (ASVs) from these heterogenous variable regions into the same reference phylogeny (Janssen et al., 2018). This enables a phylogenetically stable comparison of the microbial assemblages inhabiting these different frozen environments. In contrast to the aeolian-source microbes in englacial ice, the microorganisms in basal ice appear to represent "communities" in an ecological sense. We also used RNA-

and ATP-based approaches to identify potentially active taxa and regions in basal ice types and discuss the implications of microbial processes that occur in an important transient environment of the cryosphere.

## 2. Materials and Methods

### 2.1 Ice sampling

#### 2.1.1 Taylor Glacier

Taylor Glacier is a cold-based glacier (i.e., has basal ice that is constantly below the pressure-melting point) that is an outlet of the East Antarctic Ice Sheet, terminating at the western end of Taylor Valley (McMurdo Dry Valleys, Victoria Land, Antarctica). During the austral summers of 2007 and 2009, two tunnels were excavated into the basal zone of the northern lateral margin of the glacier (Fig. S1). Tunnel construction was initiated on fresh ice aprons and extended ~8 m into the glacier interior. At the end of each tunnel, a vertical shaft (2007 expedition) or large chamber (2009 expedition) was

excavated to expose a stratigraphic sequence of multiple basal ice types (Montross et al., 2014). Electric chainsaws equipped with unlubricated carbide tipped chains were used to exhume samples of the ice sequence from each basal ice type. Ice samples were sealed in polypropylene flat wrap, packed into insulated containers, and shipped frozen from the field to Louisiana State University where they were subsequently stored in a walk-in freezer set to -20 °C. Monitoring indicated an actual temperature of -18.9 ± 1.03 °C (n=6800) was maintained during storage.

#### 2.1.2 Matanuska Glacier

Matanuska Glacier is a 43 km temperate (i.e., englacial and basal ice constantly at the pressure melting point), valley glacier located in south-central Alaska, approximately 140 km northeast of Anchorage. In July 2013, a horizon of debris-rich basal ice exposed near the glacier terminus was sampled using an electric chainsaw (Fig. S2). After collection, the ice samples were packaged in polypropylene flat wrap and shipped frozen to Louisiana State University for storage at -20 °C.



## 2.2 Ice decontamination

Between August and September of 2013, within a temperature-controlled environmental room (-5°C), ~2 kg of ice was cut from larger basal ice samples using either a band saw (for the sediment-poor basal ice) or a diamond-bladed masonry saw (sediment-rich basal ice). The surface of freshly cut samples were then cleaned following previous methods (Christner et al., 2005). Briefly, the outermost surface was physically removed by scraping with an autoclaved microtome blade and the newly exposed ice surface was then washed with chilled (-5°C), 0.22 µm filtered 95% ethanol. Microtome scraping was omitted when processing sediment-rich samples due to the presence of coarse granules and stones embedded in the sediment-layers. Following the ethanol wash, samples were washed extensively with ice-cold, 0.22 µm, twice-autoclaved deionized water until a minimum of ~5 mm of the outer surface had been removed based on the ice weight. Large, sterilized surgical forceps were used to handle all ice samples during decontamination and were exchanged frequently during the procedure to prevent carryover contamination. Sample weights before and after decontamination were used to estimate the amount of material removed from each sample during processing, which was typically between 15% and 25% of the initial mass.

The effectiveness of the decontamination procedure was monitored by coating ice samples beforehand with a tracer solution that consisted of three components: [1] ~$10^8$ cells $mL^{-1}$ of *E. coli* JM109 cells transformed with a pETBlue-2 plasmid (Novagen) containing the gene for alcohol dehydrogenase (ADH) from *Drosophila melanogaster*, [2] fluorescein (1000 ppm), and [3] 33% (v/v) glycerol to prevent freezing on the ice at -5ºC. The removal of the tracer was monitored after each step of the decontamination procedure (i.e., scraping, ethanol wash, $H_2O$ wash). The fluorescein fluorophore component was monitored by quantifying blue-green fluorescence in 200 µL of rinse-water or meltwater using a BioTurner 20/20n Luminometer (P/N 2030-002) equipped with the blue fluorescence module (BioTurner 2030-041). The pETBlue-2 plasmid component was monitored using a PCR with the following conditions: 1 U of Taq DNA polymerase (5PRIME), 1× MasterTaq buffer, 1× TaqMaster PCR enhancer, 1.5 mM $Mg^{2+}$, 0.2 µM of each primer (TF7: 5'-TAATACGACTCACTATAGGG -3'; pETBlue-DOWN: 5'-GTTAAATTGCTAACGCAGTCA-3'), 200 µM dNTPs, and ~100 pg of template DNA. Thirty cycles of PCR were done with a 30 s denaturation step at 94 °C, 60 s annealing step at 55 °C and extension at 72 °C for 60 s, followed by a final extension at 72 °C for 10 min. The presence of viable *E. coli* was monitored by spread plating rinse-water and meltwater on agar-solidified LB media containing ampicillin (100 µg $mL^{-1}$) followed by overnight incubation at 37 °C. Samples in which any component of the tracer solution was detected in the final meltwater were discarded and reprocessed from a new subsample.

## 2.3 ATP analysis

Immediately after melting, samples were filtered through 0.22 µm pore size polyethersulfone (Supor; Pall) filters using a bench-top vacuum manifold (≤ 20 kPa). Clean ice samples were filtered through a 25 mm diameter filter while banded, solid,



and dispersed ice samples were processed through either a 47 mm or 90 mm diameter filter, depending on total sediment content. The filters were cut into small pieces using a sterile scalpel and extracted by vortexing for 10 s in 5 mL of boiling Tris-acetate buffer (TAB; 50mM Tris base adjusted to pH 7.75 with acetic acid). The mixture was then incubated in a boiling water bath for 5 minutes and immediately placed on ice for 30 minutes. Following centrifugation for 5 minutes at 4500×g to remove debris, the supernatant was collected.

The concentration of extracted ATP was measured using a modification of the firefly luciferase-luciferin assay described by (Amato and Christner, 2009). Briefly, 10 µL of extract was added to an equal volume of triethanolamine buffer (TEOA buffer; 200 mM triethanolamine pH 7.6, 2 mM MgCl2, 240 mM KCL) and incubated at 37°C for 10 minutes. To correct for adenylate adsorption to and luciferase inhibition by the sediment in the sample extracts, two samples were prepared in parallel for each measurement. Following incubation at 37°C, 10 uL of TAB supplemented with 10 µM ATP was added to one of the samples, providing an internal standard for each measurement. All samples were kept on ice during preparation and were equilibrated to room temperature (22 °C) prior to the addition of luciferase and measurement of luminescence.

Weighted linear regression was used to generate calibration curves using concentrations of ATP (in TAB) from 1 pM to 10 µM. The luciferase-luciferin cocktail was prepared fresh for each series of measurements and consisted of 100 U ml$^{-1}$ luciferase and 0.140 mM D-luciferin. Luminescence was quantified with a 20/20n luminometer (Promega) using auto-injection of 100 µL of the luciferase-luciferin cocktail. Relative luminescence units (RLU) were integrated for three seconds immediately after injection. Sample RLU values were corrected for adsorption and inhibition using the following formula:

$$\frac{1}{\left(\frac{RLU_{\text{sample+std}} - RLU_{\text{sample}}}{RLU_{\text{blank+std}}}\right)} \times RLU_{\text{sample}} = RLU_{\text{corrected}}$$

The corrected RLU values were used together with the standard calibration curve to determine the ATP concentration in the extractant. Final ATP concentrations were corrected for dilution and divided by the mass of the ice sampled. All values reported are the means of replicate samples (n = 3 to 5).

**2.4 Nucleic acid extraction**

Decontaminated basal ice samples were placed in sterilized containers and melted at 4°C. All samples except one melted in less than 24 hours and were immediately processed. The remaining sample, a large 8.5kg clean ice sample for RNA extraction (see below), took 72 hours to melt. For samples from sediment-rich basal ice (i.e., banded, solid, and dispersed ice), the resulting meltwater was shaken to resuspend any sediment and DNA was extracted from the resulting meltwater-sediment slurry using MoBio PowerMax DNA Isolation kits. To increase DNA extraction yields, two 10g extractions were combined onto a single silica spin column for each sample. For samples from sediment-poor basal ice (i.e., clean ice), meltwater was filtered onto 47 mm, 0.22 µm pore size Supor PES filter membranes (Pall) with ≤20 cm Hg of vacuum. DNA



was extracted from the filters using MoBio PowerWater DNA Isolation kits as per the manufacturer's instructions with one exception: the mechanical cell lysis step was performed on a BioSpec MiniBeadbeater-8 for 2 minutes at maximum speed.
All DNA samples were stored at -80°C until PCR amplification.

Due to the low cell abundances in the basal ice ($\leq 10^4$ cells g$^{-1}$; (Doyle et al., 2013)), very large samples (~5 kg of ice; Table S1A) were processed to recover sufficient RNA for reverse transcription PCR (RT-PCR). To facilitate filtration of the large sample sizes, coarse sediment was removed from sediment-rich basal ice samples after melting using low speed
centrifugation (700×g, 10 min, 4°C). The resulting supernatant containing fine clay and silt-like particles was then concentrated onto 90 mm, 0.22 μm pore size Supor PES filter membranes. Meltwater from clean ice samples did not require this centrifugation step and were directly filtered onto 90mm, 0.22 μm pore size Supor PES filters.

After filtration, the filters were immediately processed for RNA extraction using a modified phenol-chloroform extraction
protocol (Dieser et al., 2014). Briefly, filters were sliced into small pieces using a sterile scalpel, immersed in 3 mL of TE buffer (1 mM EDTA, 10 mM Tris; pH 6.3) containing lysozyme (15 mg mL$^{-1}$), and vortexed for 30 minutes at room temperature. Samples were then homogenized in a BioSpec Mini-Beadbeater for 2 minutes at maximum speed using ~1 g of sterilized 0.1 mm diameter zirconia/silica beads. After bead beating, crude extracts were amended with two volumes of chilled (4°C) denaturing buffer (4M guanidine thiocyanate, 50 mM Tris, 10 mM EDTA, 1% w/v N-lauroylsarcosine, 1% β-
mercaptoethanol), the resulting insoluble material was pelleted via centrifugation (4500×g for 5 min at 4°C), and the supernatant was collected. The pellet was washed with 3 mL of chilled denaturing buffer, centrifuged again, and the resulting supernatant pooled with the first. This pooled lysate was then extracted with an equal volume of phenol:chloroform:isoamyl alcohol (25:24:1, pH 6.6), followed by a second extraction with chloroform:isoamyl alcohol (24:1). Nucleic acids were purified from these extracts via an overnight ethanol precipitation with 0.3 M sodium acetate (pH
6.0). Linear acrylamide (20 μg mL$^{-1}$; Ambion) was used as a coprecipitant to aid recovery. Genomic DNA was eliminated from RNA extracts by digesting for one hour at 37°C with 4 U of TURBO DNAase (Ambion) followed by purification with MEGAclear Transcription Clean-Up kits (Ambion). Extracted DNA and RNA concentrations were measured using Quant-it Picogreen and Ribogreen kits (Life Technologies), respectively, per the manufacturer's instructions.

**2.5 16S rRNA amplicon sequencing**

From the extracted RNA, complementary DNA (cDNA) was reverse transcribed from ~1 ng of total RNA using SuperScript II reverse transcriptase with the 806R primer (5'-GGACTACVSGGGTATCTAAT-3') following the manufacturer's protocol. Controls lacking RNA template or reverse transcriptase were analyzed to monitor for contaminating RNA and the persistence of genomic DNA, respectively, in the samples.



The hyper-variable V4 region of the 16S rRNA gene was PCR amplified from the DNA extracts and cDNA libraries using a barcoded 515F-806R primer pair (Caporaso et al., 2012). Each 50 µL PCR contained ~100 pg of DNA or cDNA, 2.5 U of AmpliTaq Gold DNA polymerase LD (Invitrogen), 1× Gold Buffer, 2.5 mM MgCl$_2$, 0.2 µM of each primer, and 0.2 µM dNTPs. Amplification conditions included at 95°C for 9 min, followed by 30 to 40 cycles of denaturation at 94°C for 1 min, annealing at 50°C for 30 s, and extension at 72°C for 30 s. A final extension was performed at 72°C for 10 min.


Following amplification, amplicons were separated by electrophoresis on a 2% agarose gel to assess relative band intensity and size. The amplicons were quantified with Quant-it fluorometry and pooled at equimolar concentrations. Two extraction blanks were included in the pooled libraries to serve as procedural controls (Salter et al., 2014; Eisenhofer et al., 2019). Pooled libraries were purified with a MoBio UltraClean PCR Cleanup Kit and sequenced on the Illumina MiSeq platform

(v2 chemistry, 2× 250 bp; Georgia Genomics Facility).

**2.6 16S rRNA amplicon analysis**

Sequence read curation and processing was performed using DADA2 (Callahan et al., 2016) with the following filtering parameters (maxN=0, trunQ=2, rm.phix=TRUE, maxEE=2, R1 truncLen=240, R2 truncLen=200). Error rates for the filtered and trimmed R1 and R2 reads were calculated using the *learnErrors* function and subsequently used to denoise reads using

the DADA2 sample inference algorithm. The denoised reads were merged into amplicon sequence variants (ASV) using a global ends-free alignment. Paired reads containing mismatches in the overlapping region were removed from the dataset. Chimeric ASVs were identified and removed by using the consensus method within the *removeBimeraDenovo* function. A consensus taxonomy for each ASV was then assigned using a naïve Bayesian classifier (Wang et al., 2007) trained on release 132 of the SILVA reference database (Quast et al., 2013). The dataset was then subsampled to an even read depth across all

samples (59,562 sequences). Any ASV classified as a chloroplast, mitochondria, or from which ≥0.1% of its reads were from a procedural blank were removed (Table S1B). ASVs observed exclusively in the RNA libraries were flagged as a potential contaminant and removed from the analysis (Table S1C).

**2.7 Meta-analyses of microbiomes of perennially frozen ecosystems**

Publicly available datasets on the NCBI Sequence Read Archive (SRA) from perennially freshwater frozen environments

were identified using a search on August 8[th], 2021 with the following syntax: ice metagenome[Organism], glacier metagenome[Organism], and permafrost metagenome[Organism]. Run tables from unambiguously labeled 16S rRNA amplicon datasets were aggregated and studies based on Sanger sequencing were excluded; only datasets based on Illumina, 454, or IonTorrent sequencing were considered in the analysis. Because many studies contained samples from laboratory experiments or field manipulations (e.g., nutrient amendment, thawing, etc.), each dataset was manually curated and cross-

referenced with a publication or other data source so that only high quality 16S rRNA surveys from genuine, unmodified permanently frozen environments were included. As part of this effort, datasets were filtered to remove samples from

environments that were only seasonally frozen. For example, samples of an active soil layer or transition zone from permafrost datasets were excluded. Additionally, because of their marine origin, we also choose to exclude the wealth of sea ice sample data available and focus the analysis on terrestrial icy environments. Unfortunately, several datasets also had to

be excluded because their sequence reads on the SRA were not demultiplexed and barcode information was not provided in the linked metadata or publication. A list of datasets that met these criteria and were included in the meta-analysis are detailed in Table S1D. All runs were downloaded using the SRA Toolbox *fastq-dump* function.

Each dataset was individually processed using DADA2 to infer ASV sequences and produce an ASV table. For each study,

the filtering parameters were adjusted to account for variation in fragment length and the 16S rRNA variable region sequenced. Due to higher rates of homopolymer and indel errors with 454 and IonTorrent sequencing, datasets using these platforms used a band size parameter of 32 and homopolymer region ($\geq$ 3 repeated bases) gap cost of -1 during the sample inference algorithm. For Illumina datasets, the band size parameter was set to 16 and gaps in homopolymer regions were treated as normal gaps (gap cost of -8). Nine datasets had to be removed from the meta-analysis after these steps due to the

loss of too many sequences during quality control steps within the DADA2 pipeline. Complete parameter details and throughput statistics for each dataset in the final meta-analysis are available in Table S1E and S1F. In datasets containing clearly labeled replicates, read counts from replicate samples were combined and averaged before downstream analyses.

ASV sequences from each study were merged and inserted into the 99% Greengenes reference tree (McDonald et al., 2012)

using the SATé-enabled phylogenetic placement (SEPP) technique (Mirarab et al., 2012) within the QIIME2 *fragment-insertion* plugin (Janssen et al., 2018). The merged ASV tables from each study were filtered to remove ASVs rejected from tree insertion by SEPP. ASVs which could not be taxonomically classified beyond the domain rank (n=2434) or were classified as mitochondrial sequences (n=119) were flagged as potential artifacts and removed.

The statistical significance of hierarchical clustering patterns between samples in the meta-analysis was performed using Monte Carlo simulations of pairwise unifrac distances as implemented in the *sigclust2* R package (Kimes et al., 2017). Because parallelized, matrix-based unifrac calculators are incompatible with *sigclust2*, we used the serial, pairwise unifrac calculator from the *scikit-bio* python package. This calculator was embedded into R for the *sigclust2* Monte Carlo simulations using the *reticulate* R package.




## 3. RESULTS

### 3.1 Description of Basal Ice Samples

Based on the nomenclature of (Hubbard et al., 2009), we identified three distinct types of basal ice in the basal ice profiles sampled from Taylor Glacier: clean ice, banded ice, and solid ice (Table 1; Fig. S4; (Montross et al., 2014)). All three types

were collected during the 2007 expedition, while only clean ice and banded ice horizons were targeted during the 2009 expedition. The basal ice recovered from Matanuska Glacier was identified as dispersed ice and was the only type observed at the location of sampling.

**Table 1. Description of basal ice types collected in this study**

| Type | Glacier | Temp (°C) | Description[c] |
|------|---------|-----------|----------------|
| Clean ice | Taylor | -15[a] | debris-free ice |
| Banded ice | Taylor | -15[a] | finely stratified layers of debris and ice, layers range in thickness from a few mm to several cm |
| Solid ice | Taylor | -15[a] | composed primarily of frozen debris with only interstitial ice, no visible layering |
| Dispersed ice | Matanuska | ~0[b] | contains scattered debris aggregated into small clusters, no visible layering |

[a]Montross *et al.,* 2014. [b]Lawson *et al.,* 1998. [c]Hubbard *et al.,* 2009

### 260  3.2 Biomass in sediment-rich and sediment-poor basal ice

The quantity of DNA and RNA extracted from the basal ices was divided by the sample mass to assess the relative amounts of microbial biomass present within each basal ice type. Approximately 10,000-fold more DNA per gram of basal ice was extractable from the sediment-rich basal ice types (i.e., banded, solid, and dispersed) than in the sediment-poor clean ice (Table 2). Assuming a bacterial genome weight of 2.5 fg, the concentration of DNA extracted from the clean Taylor Glacier

basal ice corresponds to 560 genomes per gram of ice, which agrees well with direct cell counts of samples from this ice facies (Table 2; (Doyle et al., 2013; Montross et al., 2014)). In contrast, genome abundances inferred from DNA concentrations in the sediment-rich Taylor Glacier basal ice samples were two to three orders of magnitude higher than direct cells counts. Among the sediment-rich basal ice types, the temperate dispersed basal ice from Matanuska Glacier contained approximately 6-fold more DNA per gram of ice than the banded and solid basal ices from the cold-based Taylor

Glacier (-15°C). The concentration of extractable RNA was very low in all basal ice types (between 0.7 to 3.2 pg RNA g$^{-1}$; Table 2). The highest RNA concentration—though only marginally—was observed in Matanuska Glacier's dispersed basal ice. These RNA concentrations were consistent with ATP concentration data, a proxy for viable microbial biomass, and in general, high concentrations of ATP g$^{-1}$ ice were observed in sediment-rich basal ice types versus those with low sediment content.






**Table 2. Cell counts, biomass estimates, and yield of DNA and RNA extracted from basal ice samples. Sediment-rich basal ice facies are highlighted in gray.**

| Sample | Cell Density[a] cells g⁻¹ ice | Yield (pg g⁻¹ ice) DNA | Yield (pg g⁻¹ ice) RNA | Biomass Estimates genomes[b] g⁻¹ ice | Biomass Estimates ATP (pg g⁻¹ ice) |
|---|---|---|---|---|---|
| Clean07 | $0.3 - 0.5 \times 10^3$ | 4.7 | 0.7 | $1.9 \times 10^3$ | $<0.01 - 0.04$ |
| Clean09 | n.d. | 1.4 | n.d. | $0.6 \times 10^3$ | n.d. |
| Banded07 | $1.8 - 18.2 \times 10^3$ | $1.7 \times 10^4$ | 2.3 | $6.8 \times 10^6$ | n.d. |
| Banded09 | n.d. | $1.6 \times 10^4$ | 0.8 | $6.5 \times 10^6$ | $0.24 - 2.8$ |
| Solid | $6.7 \times 10^3$ | $1.4 \times 10^4$ | 1.2 | $5.7 \times 10^6$ | $<0.01$ |
| Dispersed | n.d. | $9.9 \times 10^4$ | 3.2 | $4.0 \times 10^7$ | $0.06 - 0.14$ |

[a]Doyle et al. 2013. [b]Number of microbial genomes estimated using 2.5 fg of DNA per genome. N.d.= no data

### 3.3 Composition of microbial assemblages in different types of basal ice

A total of 4,227,678 paired-end reads with an average read length of 253 bp were obtained from MiSeq sequencing. After filtration and denoising, 2,883,221 non-chimeric sequences representing 3,310 ASVs remained. Subsampling and culling of ASVs classified as plastids or flagged as potential contaminants produced a final, curated dataset composed of 2,005 ASVs.

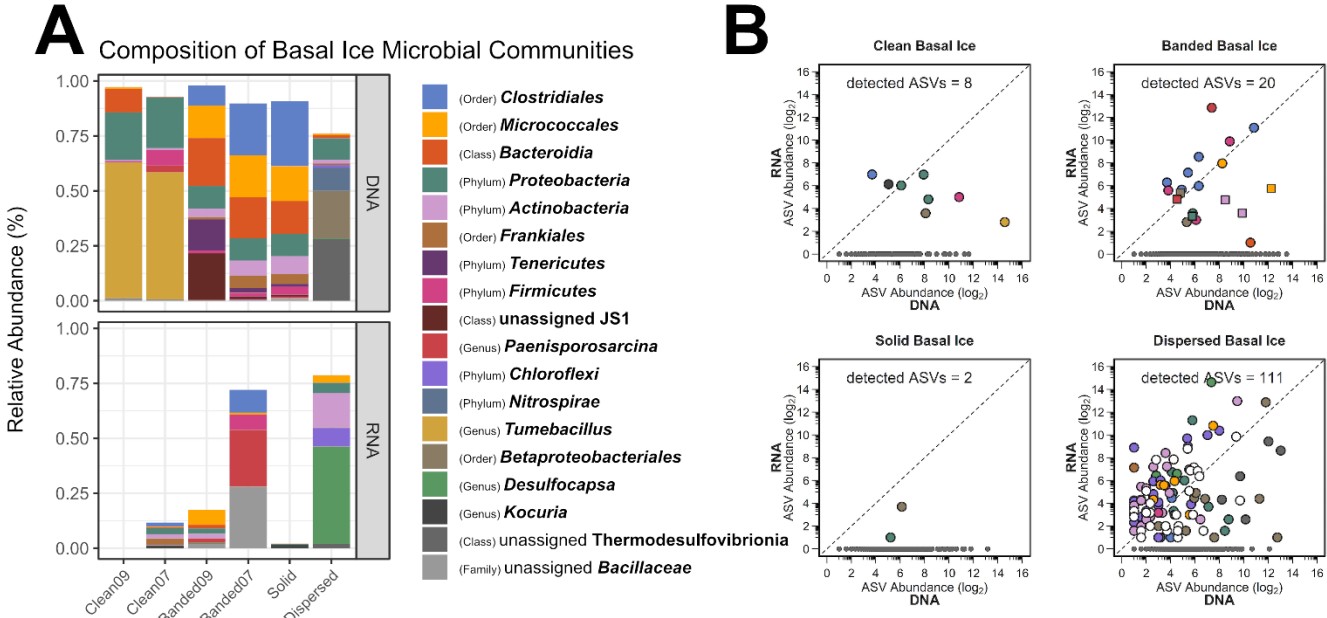

**Figure 1. [A]** Relative abundances of microbial lineages observed in basal ice based on the use of DNA and RNA templates for
detecting 16S rRNA sequences. The graph displays the highest resolution classification for the most abundant taxa and was constructed as follows: First, ASVs were clustered by genus and any having a relative abundance ≥20% in at least one of the samples were plotted. This procedure was subsequently repeated with the remaining unplotted ASVs at the rank of family, order, class, and finally phylum. Any remaining rare taxa left after this procedure were not plotted. The bottom plot was scaled to reduce inflated relative abundance biases introduced during curation. **[B]** Scatter plots of potentially active ASVs within four types of
basal ice. Each point represents an individual ASV. The dotted-line indicates a 1:1 ratio between the RNA- and DNA-based 16S rRNA libraries. Gray points denote ASVs observed only in the DNA-based libraries. In the banded basal ice plot, circles denote ASVs from the Banded07 sample while squares denote ASVs from the Banded09 sample. Detected ASVs in each plot denotes the number of ASVs observed in both the RNA- and DNA-based 16S rRNA libraries.



Based on average-neighbor clustering of Bray-Curtis distances, the basal ice samples clustered into three significantly
different (AMOVA; $F_{[2,3]}=5.23$, $p=0.017$) groups: (1) Taylor Glacier sediment-poor basal ice, (2) Taylor Glacier sediment-rich basal ice, and (3) Matanuska Glacier dispersed ice. The sediment-poor clean ice samples contained microbial assemblages overwhelmingly composed of *Firmicutes* (68%) and *Proteobacteria* (25%), with smaller proportions of *Bacteroidetes* (5%) and *Actinobacteria* (1%). The genus *Tumebacillus* was especially abundant in the clean ice samples (Fig. 1A and S3), representing more than half of all taxa in these samples. Assemblages within the banded and solid ice samples
were comparatively more diverse than those found in clean ice (Fig. 2), with a wider range of abundant phyla represented: *Actinobacteria* (26%), *Firmicutes* (23%), *Bacteroidetes* (19%), *Proteobacteria* (11%), *Atribacteria* (9%), *Tenericutes* (7%), and *Planctomycetes* (1%). Notably, ASVs related to the class *Clostridia* composed about 20% of the total in these samples. The microbial assemblage within Matanuska's dispersed basal ice was distinct from those observed from Taylor Glacier. Here we observed significant amounts of *Nitrospirae* (39%) and *Betaproteobacteria* (30%), of which several of the most
abundant ASVs were related to sulfur and/or iron cycling members of the genera *Thermodesulfovibrio*, *Rhodoferax*, and *Thiobacillus*. Comparing assemblages from all basal ice samples, only 21 taxa (~3%) were shared between all four basal ice types (Table S1G). These shared taxa were generally rare members with median relative abundances ranging between 0.03% and 0.31%.

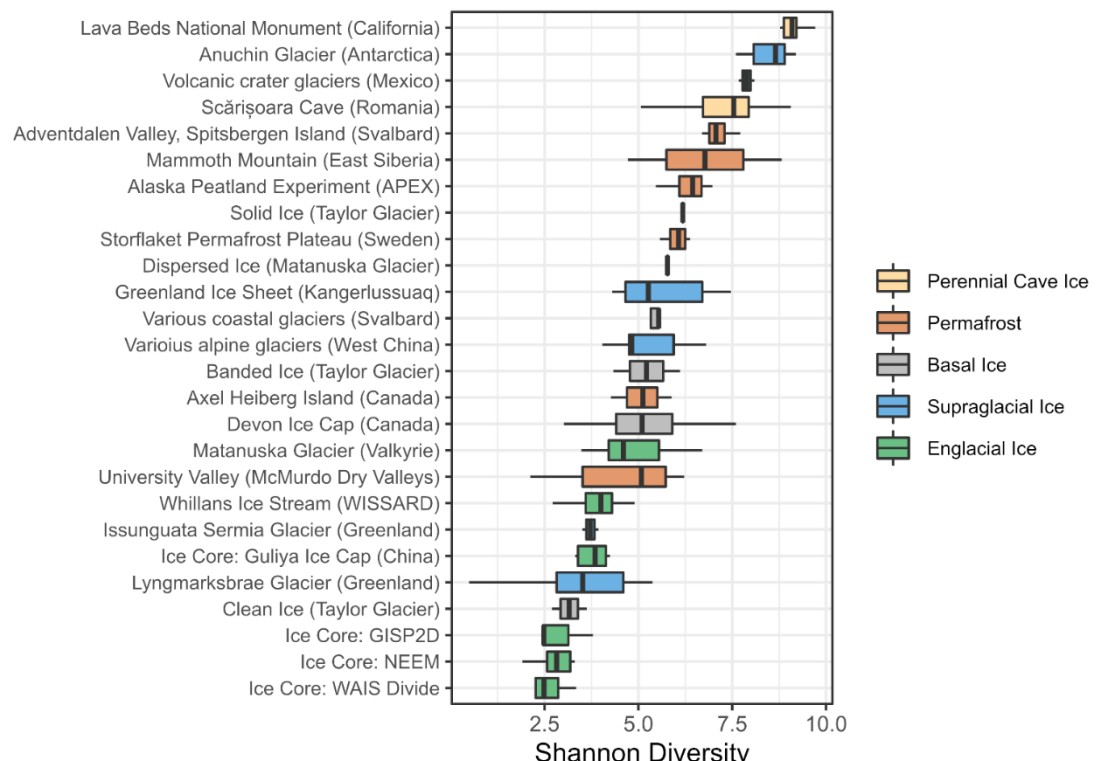


**Figure 2. Microbial alpha diversity (Shannon) within various perennially frozen environments.**





**3.4 Assessing metabolic status of bacterial taxa within different types of basal ice**

Of the 2,005 ASVs observed in this study, we detected 141 (7% of total) within the RNA-derived 16S rRNA libraries (Table S1H), suggesting they were recovered from potentially intact cells rather than eDNA or dead cells. From this group, 80 of the ASVs had RNA/DNA abundance log-ratios >1 and likely represent metabolically active taxa within the basal ice. The vast majority (63 ASVs) of these high-ratio ASVs were observed in the dispersed ice from Matanuska Glacier (Fig. 1B), the warmest and youngest basal ice sampled in this study. Many of the Matanuska Glacier ASVs were members of the *Chloroflexi* and *Deltaproteobacteria,* with some of the largest RNA/DNA abundance log-ratios belonging to *Desulfocapsa*, *Syntrophus*, and members of the family *Anaerolineaceae*. Within the banded basal ice from Taylor Glacier, over half of ASVs detected in the RNA-derived libraries were members of the *Firmicutes*. Among this phylum, members of the *Paenisporosarcina*, *Paenibacillaceae*, *Desulfosporosinus*, *Caldicoprobacter*, and *Virgibacillus* had the highest activity potential based on RNA/DNA abundance log-ratios (Table S1H) and represented >90% of the RNA reads in these samples. In comparison, the clean and solid basal ice samples contained the least number of potentially active members—10 ASVs from only 472 RNA total reads, only 2 of which had RNA/DNA abundance log-ratios >1.

**3.5 Meta-Analysis: Microbial populations of various perennially frozen environments**

For the meta-analysis, we analyzed 54.6 million 16S rRNA sequences across 256 individual samples from 24 publicly available datasets that represent five different types of freshwater ice environments: (1) supraglacial ice, (2) englacial ice, (3) basal ice, (4) perennial cave ice, and (6) permafrost. After sequence filtering and curation, 7 samples were excluded from meta-analysis due to low read depth. The final meta-analysis dataset contained 25.7 million non-chimeric sequences representing 54,170 unique ASVs. Subsampling to equal read depth reduced this to 35,357 ASVs.

We calculated Shannon diversity indices to compare microbial alpha diversity between these frozen environments. Within the basal ice assemblages characterized in this study, alpha diversity was significantly higher in the sediment-rich basal ice types than the low sediment clean ices ($t(4)=3.34$, $p=0.03$) (Fig. 2). Compared to the other frozen environments in the meta-analysis, alpha diversity within Antarctic sediment-rich basal ice samples was most like that observed in basal ice from various coastal glaciers in Svalbard (Tukey HSD mean difference = 0.07, $p=1$) (Perini et al., 2019). The lower alpha diversity values within Taylor Glacier's clean ice were most like those found in englacial ice samples from the GISP2D, NEEM, Guliya Ice Cap, and WAIS Divide ice cores (Tukey HSD mean difference = 0.39, $p=1$) (Miteva et al., 2015, 2016; Price et al., 2015; Zhong et al., 2021).

To parameterize microbial β-diversity between samples, we used hierarchical cluster analyses of both unweighted (UF) and weighted UniFrac (wUF) distances (Fig. 3 and Fig. S5). In the UF cluster analysis (Fig. 3), microbial communities largely clustered by environment type, indicating presence/absence data alone can reasonably discriminate between different parts of





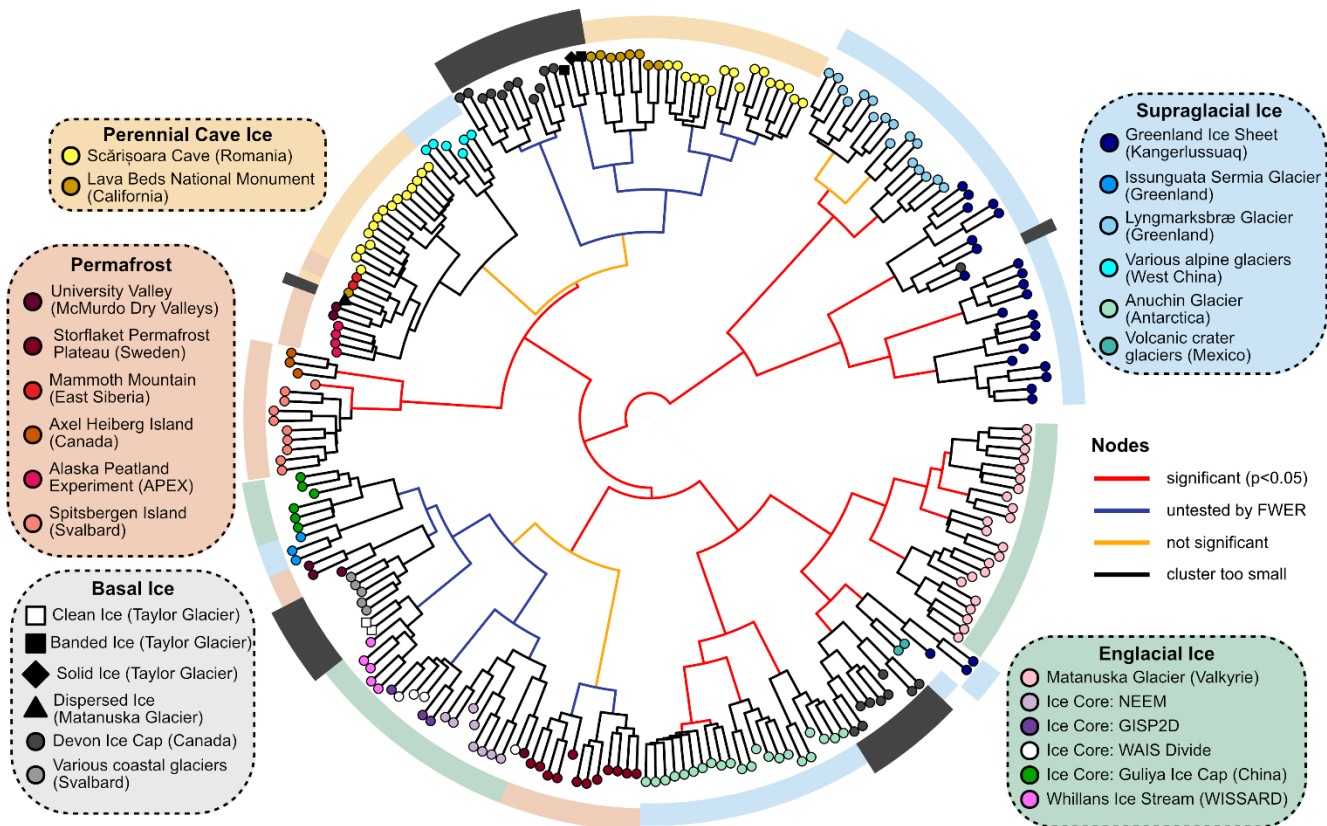

**Figure 3. Hierarchical cluster analysis of UF distances between samples using Ward's minimum variance method. The significance of each node was calculated with a Monte Carlo simulation using the 'sigclust2' R package. Family-wise error rate (FWER) was controlled across the dendrogram using a sequential testing procedure starting at the central node.**

the cryosphere. This suggests microbial community composition in permanently frozen environments is strongly influenced by founder effects, possibly due to a population bottleneck effect occurring upon freezing. In the cluster analysis of wUF, the overall cluster pattern was comparable but many of the branching patterns between samples were more statistically significant, indicating the inclusion of abundance data can help discriminate samples from the same type of environment.

Within the cluster analyses, we also found some samples displayed larger variation and overlap in community structures than others. For example, microbial assemblages within englacial ice were the most similar (UF: $0.47 \pm 0.15$; mean ± s.d; n = 42). In contrast, perennial cave ice samples (Itcus et al., 2018; Paun et al., 2019) had the highest heterogeneity among samples (UF: $0.85 \pm 0.59$; n=14). β-diversity of permafrost microbiomes were intermediate of these endmembers, with relatively tight clustering observed in permafrost from the Storflaket peat bog (northern Sweden) but wider variation in permafrost samples from University Valley (Antarctica) (Goordial et al., 2016; Monteux et al., 2018). The microbial communities within Taylor Glacier's banded and solid basal ice facies clustered closely with perennial cave ices located in both Romania (Itcus et al., 2018; Paun et al., 2019) and California's Lava Bed National Monument (O'Connor et al., 2021). Those within the clean





basal ice were instead most similar to assemblages observed in englacial ice from the NEEM and GISP2D ice cores from Greenland (Miteva et al., 2015, 2016)(Fig. 3).

We used an RDA model to determine how microbial lineages were distributed across the different types of frozen
environments (Fig. 4, left). The final RDA model explained 50.7% of the variability between samples and effectively separated the samples into three clusters, echoing patterns observed in the unifrac cluster analysis (Fig. 3). Of the 1627 microbial lineages included in the analysis, we identified 36 with loading vectors greater than the equilibrium contribution (i.e., the proportion of variance that would be explained by a random constrained axis), indicating they were associated with one or more of the five types of frozen environments. Scalar projections of these lineages onto the centroid factor of each
type of frozen environment allowed us to estimate how strongly these lineages were associated with each environment (Fig. 4, right). Unsurprisingly, many supraglacial ice environments harbored comparatively higher proportions of *Cyanobacteria* and plastid-harboring taxa than the other frozen environments. Englacial ice environments were enriched with members of the genera *Pseudomonas*, *Methylobacterium*, *Acinetobacter*, and *Polaromonas*, *Massilia*, and *Janthinobacterium*. In contrast, basal ice, permafrost, and perennial cave ice environments all contained higher membership of *Firmicutes* and
*Actinobacteria* related taxa such as *Clostridium*, *Desulfosporosinus*, *Oryzihumus*, and *Cryobacterium*.

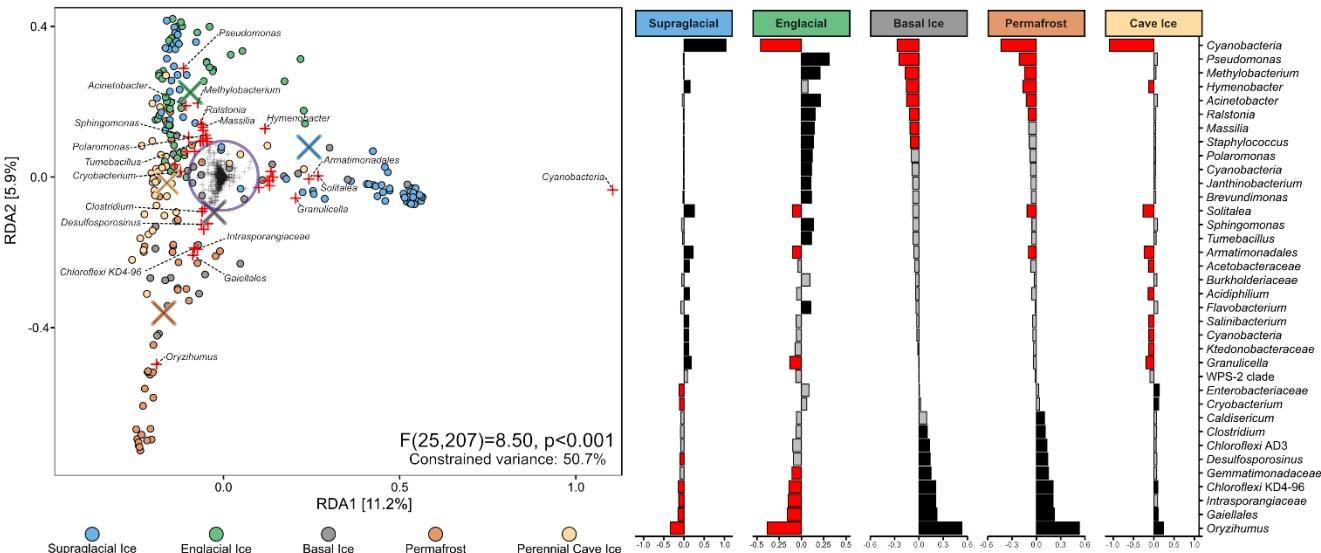

**Figure 4. [Left] RDA triplot of the distribution of 1627 microbial lineages across five different perennially frozen environments. Colored circles represent individual samples. Centroids of each environment type are indicated with an "×". Plus signs (+)**
**represent microbial lineages. The purple circle denotes the equilibrium contribution of the overall model and delineates lineages which substantially constrained the overall ordination. Those lineages are highlighted in red. [Right] Magnitude of the scalar projections of these 36 lineages onto the centroids for each ice types approximates that lineage's association with that frozen environment. Scalar projections smaller than the equilibrium contribution were considered inconclusive and marked gray. Positive and negative associations are black and red, respectively.**



### 3.6 Meta-Analysis of early studies based on 16S rRNA clone sequencing

Although 16S rRNA clone libraries lack the read depth to include in our primary meta-analysis, we performed a secondary analysis to compare how many of the microbial taxa observed in several early cryosphere microbiome studies (Yde et al., 2010; Skidmore et al., 2005; Cheng and Foght, 2007; Katayama et al., 2007; Steven et al., 2008) compared to the 'indicator' lineages we identified in Fig. 4. To achieve this, we inserted full-length clone sequences into the meta-analysis reference tree using SEPP and taxonomically classified them using Wang's Bayesian inference method with the Silva v.132 database as a reference. Clone sequences were then agglomerated together with meta-analysis ASVs belonging to the same genera.

**Figure 5. Distance-based redundancy analysis of Aitchison distances between different perennially frozen environments. Colored circles represent centroids of each study's samples. Black triangles represent individual early studies of basal ice and permafrost based on Sanger sequencing of 16S rRNA clone libraries.**

Of the 236 16S rRNA clone sequences included in this secondary analysis, all but three belonged to a taxon that had been observed in the meta-analysis. Despite their extremely low read depth (between 11 and 136 clones), these five studies clustered with similar environments in a distance-based RDA ordination (Fig. 5). Remarkably, of the 36 indicator taxa we



identified in the primary meta-analysis, 12 were represented by at least one clone sequence among these five studies. Many

of these were among the most abundant sequences observed in these clone libraries (Fig. S6).

## 4. DISCUSSION

### 4.1 Microbial communities vary in size and composition across basal ice types

Of the four different types of basal ice analyzed in this study, we found that those containing large quantities of sediment or debris contained higher quantities of extractable nucleic acids and microbial abundances than those containing low sediment

content. This is consistent with previous microbiological investigations of the GISP2 ice core that found higher cell concentrations in the deepest sections of the core, where clay particles were embedded in the basal ice (Tung et al., 2006). In addition to higher numbers of total cells and biomass, microbial diversity is also notably higher in the sediment-rich basal ice facies relative to clean basal ice or englacial ice, implicating subglacial debris as the primary source of microorganisms in the basal ice.


Our estimates of bacterial cell abundance based on the yield of extracted DNA (~$10^6$ cells g$^{-1}$ ice) were approximately 100-fold higher than those based on direct microscopic counts (~$10^4$ cells g$^{-1}$ ice) in the sediment-rich banded and solid basal ice. However, in the clean basal ice samples, these estimates were highly comparable (Table 2). This difference likely reflects the much greater efficiency of DNA extraction versus that of cell extraction. It may also be related to the increased prevalence of

eukaryotic genomes or environmental DNA that is associated with sediment-rich basal ices (Perini et al., 2019; Sonjak et al., 2006; Fraser et al., 2018). Alternatively, Taylor Glacier's debris-rich basal ice may contain significant amounts of necromass. This would be consistent with previous studies that have found some deep, debris-rich horizons of basal ice, such as those of the Greenland ice sheet, contain ancient DNA from buried organisms (Willerslev et al., 2007).

We found microbial assemblages within basal ices of the cold-based (-15°C) Taylor Glacier to contain high abundances of *Firmicutes* and *Actinobacteria* relative to the other frozen habitats examined. These Gram-positive phyla are commonly observed as the dominant taxa in a wide range of frozen environments including ground ice, permafrost, polar saline springs, cryopegs, and sea ice (Steven et al., 2008; Kochkina et al., 2001; Steven et al., 2007; Perreault et al., 2008; Lacelle et al., 2011; Boetius et al., 2015; Spirina et al., 2017). One possible explanation for these high abundances is that the

physiochemical stresses faced by cells under frozen conditions (Doyle et al., 2012) favor the prevalence of microorganisms that can form highly resistant endospores (*Firmicutes*) or spores (*Actinobacteria*). Indeed, the ability to enter a metabolically dormant state and form an environmentally resistant spore likely represents a robust survival strategy for microorganisms that become entrapped inside ice for extended timeframes (Filippidou et al., 2016). However, multiple lines of evidence indicate this hypothesis does not adequately explain the abundance of these spore-forming taxa in our basal ice samples.

First, ASVs related to known endospore-forming genera such as *Paenisporosarcina*, *Virgibacillus*, *Clostridium*, and



*Desulfosporosinus* were abundant in all our RNA libraries. Because the rRNA content of endospores decreases relatively quickly after sporulation—on the order of days to weeks (Segev et al., 2012; Korza et al., 2016)—their detection in >10,000-year-old basal ice implies these taxa existed as vegetative cells and not as endospores within the ice. Second, populations of *Paenisporosarcina* sp. isolated from the Taylor Glacier banded ice do not sporulate after being refrozen in basal meltwater at in situ temperatures (-15°C), but instead continuously incorporate radiolabeled DNA and protein precursors into macromolecules (Doyle et al., 2013). Third, we found pasteurization of banded basal ice meltwater dramatically reduced isolate cultivability, indicating isolates were not being recovered from heat-resistant endospores. Finally, 16S rRNA surveys of permafrost that have selectively depleted DNA from dead or vegetative cells have found many endospore-forming taxa, especially members of the *Clostridia*, were more likely to persist as vegetative cells rather than endospores in the permafrost (Burkert et al., 2019). Collectively, these results suggest the high abundances of *Firmicutes* and *Actinobacteria* in Taylor Glacier's basal ice may not only be related to their environmental durability while dormant, but also due to properties associated with their vegetative cycle that promote physiological activities within frozen matrices (Doyle et al., 2013).

**4.2 Young and warm versus old and cold basal ice**

Due to the chemical and biological lability of RNA, paired RNA/DNA 16S rRNA amplicon libraries provide a useful proxy for discerning viable and metabolically active microorganisms in a community from those that are dead or dormant. However, there are limitations to this analysis to be mindful of. Variations in sequencing depth, PCR primer bias, differences in DNA/RNA extraction, and inconsistent relationships between rRNA content and activity for different taxa can introduce biases in 16S ratios. As such, direct comparisons of 16S ratios as reliable indicators of the relative metabolic activity between different ASVs should be performed with caution (Blazewicz et al., 2013). This may especially be the case in frozen environments, where RNA may be more stable in dormant cells. As such, we used a conservative approach that focuses primarily on the number of detected taxa in each basal ice type, especially those with unambiguously high RNA/DNA abundance log-ratios, which are difficult to explain in ancient ice samples without invoking metabolic activity.

On average, only about 2% of the total ASVs in each sample were also detected in the paired RNA library. This low proportion of overlap suggests that only a small fraction of the microorganisms preserved in the basal ice are potentially metabolically active within the ice matrix, although it is possible some may have been undetected due to primer bias during reverse transcription. We took care to extract nucleic acids from samples as soon as melting was complete (typically between 12 and 24 h) to limit any impacts the melting process had on which ASVs we detected. Although one sample took much longer to melt (i.e. Clean07 for RNA), we ultimately did not find much evidence for active ASVs in this sample, suggesting the effects of the extended melting time (72 h) for this sample were minimal. The dispersed basal ice from Matanuska Glacier contained a much larger number of high-ratio ASVs than any of the basal ice facies recovered from Taylor Glacier (Fig. 1 and S3). One plausible explanation for this observation is the substantial difference in underlying geology, age, and temperature of basal ice between the two glaciers.



Assuming similar starting inputs of microbes, older ice would be expected to harbor smaller numbers of potentially active populations than younger ice due to microbial attrition over time (Doyle et al., 2012). Analysis of the δD and δ¹⁸O composition of meltwater from Taylor Glacier surface ice samples estimate ages of 11,500 to 65,000 years BP (Aciego et al., 2007), while recent radiometric ⁸¹Kr dating efforts have yielded age estimates near the glacial terminus of 123,500 years BP (Buizert et al., 2014). In comparison, measurements of oxygen isotopes in Matanuska Glacier's dispersed basal ice layer

indicated the basal ice originated in the accumulation area (Lawson and Kulla, 1978). Based on the velocity of Matanuska Glacier (~110 m yr⁻¹), this would represent an approximate time of 250 years between deposition and terminal ablation. Trace amounts of anthropogenic tritium produced by atmospheric thermonuclear weapon tests in the 1950s and 1960s have also been detected in Matanuska Glacier's basal ice zone (Strasser et al., 1996), indicating portions of this ice were formed even more recently by freeze-on of surface-derived meltwater (Lawson, 1979). This indicates the dispersed ice we sampled

from Matanuska is substantially younger than any of the basal ice types collected from Taylor Glacier, with the latter probably being between ~100- to 500-times older than the former.

In addition to age of the basal ice, the large difference in ice temperature between a temperate and cold-based glacier would also affect the potential for metabolic activity in the ice habitat (Price and Sowers, 2004; Price, 2000). Taylor Glacier is a

cold-based glacier with a basal ice temperature between -15°C and -17°C based on independent measurements (Montross et al., 2014; Samyn et al., 2008). In contrast, the Matanuska Glacier is temperate, and the basal ice zone is at the pressure melting point of 0°C (Lawson et al., 1998). As such, cells present in the liquid habitat at grain boundaries (Price, 2000) in Taylor Glacier's basal ice are under considerably higher physiochemical stress than ice near the melting point. For example, the predicted ionic strength of the briny liquid at ice crystal boundaries—where the microorganisms are located— is nearly

4.6M at -15°C. Comparatively, this is 11.5-fold higher than estimates for unfrozen water in glacial ice at -1°C (Doyle et al., 2012). Hence, the wider diversity and increased number of ASVs with a high potential for metabolic activity in the dispersed basal ice may reflect not only the sample age, but the more favorable conditions for metabolism in temperate versus cold-based glacial ice matrices.

### 4.3 Debris-rich basal ice types are hot spots for microbial activity

Multiple studies have reported observations of very low concentrations of oxygen (Souchez et al., 1995) and large excesses of $CH_4$ in debris-rich basal ice from the Greenland and Antarctica ice sheets (Tung et al., 2006; Wadham et al., 2012; Rhodes et al., 2013; Lamarche-Gagnon et al., 2019; Lee et al., 2020; Stibal et al., 2012). Mirroring these findings, an analysis of gasses entrapped within Taylor Glacier banded basal ice found $O_2$ was depleted to as low as 4% in horizons where $CO_2$ concentrations were concurrently enriched as high as 20,000-fold relative to atmospheric concentrations (Montross et al.,

2014). Isotopic analysis of $\delta^{13}C\text{-}CO_2$ revealed this $CO_2$ was isotopically depleted (-24‰), and therefore, was likely produced by microbial respiration of organic matter (Montross, 2012). In this study, among all the RNA-based 16S rRNA libraries we



sequenced from Taylor Glacier basal ice, we found the largest reads counts and largest number of potentially active ASVs in these same banded basal ice horizons. Together with elevated ATP concentrations, these observations collectively suggest Taylor Glacier's banded basal ice horizons harbor metabolically active microbiomes that alter entrapped gas compositions by actively metabolizing organic matter to $CO_2$ at -15°C within the ice matrix.

Alternatively, these trends in gas data could be the result of microbial activity that had occurred before the basal ice formed. One way to explore this possibility is by considering the total gas content of the ice. If the gases were dissolved in the water that froze onto the base of the glacier, then total gas volumes as low as 0.02 to 0.08 cc/g would be expected (Cuffey et al., 2000; Sleewaegen et al., 2003). In solid facies of the Taylor Glacier basal ice profile, the lower total gas volumes observed were indeed low (0.01 to 0.07 cc/g; (Montross et al., 2014)). Hence, although we did not see much evidence for microbial activity in the solid basal ice, we cannot exclude largescale melting/refreezing processes occurred during its formation. In contrast, in the banded basal ice—where isotopic, ATP, and RNA data provided strong evidence for microbial activity—the total gas volumes were consistent with those observed in the clean ice (0.09 to 0.11 cc/g; (Montross et al., 2014)). These volumes are more similar to those typically seen in englacial ice of meteoric origin (0.1 cc/g; (Paterson, 1994)). In summary, large-scale melting/refreezing cannot explain the gas data observed across all facies, and in the banded ice, in situ microbial respiration is the only explanation consistent with its total gas content.

This metabolic activity may simply be a consequence of substrate availability for microbial consumption. Previous geochemical analysis of Taylor Glacier's basal ice found dissolved organic carbon (DOC) concentrations are much higher in the debris-rich basal ice types (~63 mg L[-1]) than in the debris-poor types (~0.8 mg L[-1]; (Montross, 2012; Montross et al., 2014)) However, our comparison of multiple basal ice types suggests there are likely additional, unidentified factors besides sediment content controlling whether basal ice harbors an active microbiome. For example, although the banded and solid basal ice samples harbored very similar microbial communities in our DNA-based libraries, we found very little evidence for active ASVs within the solid basal ice. This suggests—at least in polar glaciers containing basal ice horizons well below 0°C—elevated debris and/or DOC are not alone sufficient for supporting metabolic activity in the ice.

In many of the debris-rich basal ice types we investigated in this study, we observed an enrichment of putatively anaerobic taxa (e.g., *Clostridia*) within the same samples where $O_2$ concentrations were depleted relative to atmospheric values. One possible explanation is that the subglacial debris within these horizons were enriched with these taxa before being entrained into the basal ice. Alternatively, these patterns represent ecological shifts in microbial community composition that occur inside the ice matrix as oxygen becomes depleted. However, there is an important nuance to how these shifts probably occur. Although there is ample evidence that some microbial taxa can remain metabolically active while frozen (Doyle et al., 2013; Panikov et al., 2006; Bakermans and Skidmore, 2011; Tuorto et al., 2014; Segura et al., 2017), there is comparatively little concrete evidence that microorganisms can physically grow and divide while frozen inside ice. This suggests that





reproduction does not occur, or if it does, is imperceptibly slow under the conditions within an ice matrix. Hence, the increased relative abundance of many putative anaerobic taxa that we observed in many of the sediment rich basal ice samples likely represents a decline of strictly aerobic taxa after oxygen has become depleted rather than a genuine enrichment or outgrowth of putative anaerobes. This hypothesis may explain why enrichment cultures inoculated with

100,000-year-old Greenland basal ice became turbid substantially faster (2 to 3 weeks versus several months) when incubated anaerobically versus aerobically (Sheridan et al., 2003).

### 4.4. Basal ice diagenesis likely controls microbiome composition and structure

The microbiomes of cold-based, debris-rich basal ice horizons such as the banded and solid basal ice from Taylor Glacier were phylogenetically most like those in perennial cave ice or permafrost. These similarities are likely a reflection of the

comparable physiochemical conditions found within these environments. All three are characterized by relatively high concentrations of debris, constant subzero temperatures, and perhaps—as suggested by the co-enrichment of several putative anaerobic taxa (e.g., *Clostridium*, *Desulfosporosinus*) in some samples—depleted or decreased oxygen concentrations. In contrast, clean basal ice appears to harbor microbiomes most like those found in englacial ice. This is consistent with the notion that clean basal ice and englacial ice are diagenetically very similar. Both contain very little to no debris and are

typically formed from the accumulation and compression of snow into glacial ice. In fact, except in unique cases where new clean basal ice is formed by the accretion of subglacial lake water onto the base of a glacier, the only real difference between clean basal and englacial ice may be location. In other words, in many cases clean basal ice is simply deep englacial ice. Collectively, these findings suggest the microbiomes of basal ice environments are not simply preserved communities but rather are actively structured by the diagenesis and physiochemical nature of their habitat.

### 4.5 Meta-analysis limitations

Comparing 16S rRNA amplicon datasets from different studies is hindered by inconsistencies in the sequencing platform used and the variable region of the 16S rRNA gene that was amplified for sequencing. Using SEPP, heterogeneous ASVs from different studies are inserted into the same reference tree. This creates stable phylogenetic placements that enable integration and meta-analysis of amplicon data from multiple variable regions of the 16S rRNA gene (Janssen et al., 2018).

Nevertheless, it is important to note the limitations of this approach. First, integrating previously published 16S rRNA amplicon datasets based on DGGE profiling remains difficult as they lack sufficient read depth or taxonomic resolution to make meaningful comparisons with more modern datasets. As a result, our meta-analysis excluded many of the pioneering works that investigated the microbial ecology of glacial ice. Second, there are potential biases in our meta-analysis due to how we classified some icy environments. For example, while it is clear supraglacial ice exists on the surface, there is no

universal definition of a depth wherein glacial ice should suddenly be considered 'englacial'. Indeed, supraglacial and englacial ice would probably be more accurately portrayed as opposite ends of a vertical gradient through the body of a

glacier rather than discrete categories. This may explain some of the overlap in microbiome composition and structure we observed between supraglacial and englacial environments (Figs. 4 and 5).

Another potential source of bias in the meta-analysis to be cognizant of is how some ice samples are collected, specifically basal ice. Due to the logistical difficulty of accessing the basal ice zone of a glacier, most samples of basal ice included in the meta-analysis were not collected from inside an excavated tunnel like our samples at Taylor Glacier. Instead, most of these samples (including our sample of Matanuska's dispersed basal ice), were collected from easily accessible horizons of basal ice that were exposed at the glacial margin. These samples have been exposed to sunlight and seasonal temperature

fluctuations that are not found in the glacial interior. It is difficult to accurately predict how these conditions would affect the basal ice microbiomes. However, it might explain why, for example, phototrophic taxa (e.g., *Cyanobacteria*) are observed in several of the basal ice samples from coastal Svalbard glaciers.

### 4.6 Conclusions and Implications

Our findings show that certain microbes do not persist in a state of dormancy while frozen, and coupled with other

biogeochemical data (Doyle et al., 2013; Montross et al., 2014), indicate that the sediment-rich basal ice horizons of glaciers harbor microbial communities that actively conduct biogeochemical cycling at subzero temperatures. Although we have limited observations, our data further implies that a larger diversity of bacteria remain metabolically active in temperate basal ice found beneath nonpolar glaciers and ice sheet interiors (Lawson et al., 1998; Bell et al., 2011). Given that the basal temperature of the Antarctic and Greenland ice sheets are near or at the pressure-melting point (Gow et al., 1968), this raises

the intriguing possibility that subglacial regions of the cryosphere may be more biogeochemically active than previously thought.

Microorganisms in the basal ice of cold-based glaciers may be important biogeochemical actors. Although rates of microbial activity in these environments are extremely low and much lower than those of warmer environments (Doyle et al., 2013;

Amato and Christner, 2009; Panikov et al., 2006; Christner, 2002; Amato et al., 2010), this may be offset by the long residence time of basal ice. Microbial processing in basal ice could have important implications to material that is transported to and melts at the margin, representing an important source of dissolved organic carbon, nutrients, and trace metals to marine ecosystems (Vick-Majors et al., 2020; Hawkings et al., 2020; Rignot and Jacobs, 2002; Jung et al., 2019).

Finally, our meta-analysis demonstrates that basal ice environments harbor a diverse range of microbial communities which can resemble those found in a wide range of other icy environments. This is likely a reflection of the many different types of basal ice, which vary considerable in both physical and chemical characteristics (46). Debris-rich basal ice types, with comparatively higher concentrations of potential nutrients and redox substrates for microbial metabolism resemble other debris-rich icy environments such as permafrost and/or perennial cave ice. Likewise, clean basal ice microbial communities



are more like those found in the oligotrophic, debris-free englacial ice that makes up the bulk of a glacier's mass. Broadly, these findings indicate basal ice diagenesis plays a major role in microbiome composition and structure.

## Data availability

Sequence read data generated in this study is available on NCBI SRA under BioProject PRJNA282540.

## Author contribution

SD and BCC designed the research, SD performed the research and analyses, SD and BCC wrote the paper.

## Competing interests

The authors declare that they have no conflict of interest.

## Acknowledgements

This study was funded by an Office of Polar Programs grant (ANT-0636828) from the National Science Foundation. We
thank Amanda Achberger, Pierre Amato, Tim Brox, Lindsay Knippenberg, Scott Montross, and Mark Skidmore for multiple discussions and as vital partners in tunnel excavation and sampling in Antarctica. We also thank the staff of McMurdo station and Petroleum Helicopters International for logistical support in the field.

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
