# Peer review of "Variation in bacterial composition, diversity, and activity across different subglacial basal ice types"

_The Cryosphere, 2022_

## Author Response (AR1)

**REVIEWER #1**

This is a very well written and presented piece of research, investigating the microbial communities associated with basal ice of different thermal regimes of glacier. The aims are clearly stated, the work rigorous and placing their results within the context of other previously published work in particular a great addition to the literature.

I have some minor comments relating to the wording / writing, and one slightly less minor point about the way in which 16S rRNA sequencing was analysed. In particular, the use of a now out of date SILVA 16S rRNA database. These comments are addressed below in order they arise in the manuscript:

**Thank you**

Line 8: "providing *a* habitat"?
**Corrected.**

Line 15: This sentence could be clearer to better distinguish between the amplification of the 16S rRNA gene from extracted DNA vs RNA (via cDNA). It took me a few reads of the sentence to see the difference in the current way it is written.
**Agreed. We have revised this sentence to "...high 16S rRNA/rDNA amplicon ratios implicated...".**

Lines 162-167: If I understand correctly, the coarse and fine sediments were separated to "*facilitate filtration of the large sample sizes*" but surely this would have led to a loss of biomass associated with coarse sediment? Acknowledgement of this would be pertinent.
**Good point. We now mention this in the revision.**

Line 193: Missing word between "included" and "at 95°C..."
**Thank you. Corrected.**

Line 209: One less minor issue I have with this manuscript is the use of SILVA v132 for taxonomic assignment of 16S rRNA gene sequencing data. The more recent version - v138 - is much larger and the associated updated taxonomy has altered lineage assignments to SSU sequencing data, especially at higher taxonomic ranks (e.g. Phylum and Class level). For what reason was v132 chosen over v138? - the former has been available since 2019, and is the norm now for taxonomic assignments. How different would your taxonomic assignments be if you used the newer version?
**Using the older SILVA v132 for taxonomic assignment was an oversight and we have reclassified the 16S rRNA data using the latest version (v138.1) in the revised manuscript.**

**The differences are largely superficial: 90.2% of the ASVs have identical assignments at the genus rank using the newest database. About 35% of the ASVs have new Phylum/Class level assignments, mostly due to minor spelling or suffix changes (*Bacteroidetes* and *Actinobacteria* are now *Bacteroidota* and *Actinobacteriota*, respectively) or modifications to the names of some lineages (e.g. *Betaproteobacteriales* have been renamed to *Burkholderiales,* with no changes at lower taxonomic ranks (e.g. Family/Genus)).**

Line 239-243: It is not clear to me what reference database was used for taxonomic assignments of these previously published datasets, please clarify.
**We have reclassified the metaaanalysis with SILVA 138.1 and added this detail to the text.**

Fig 5 caption: You mention black triangles but do you mean black diamonds? I can't see any triangles on the figure
**Thank you for catching this oversight. The caption has been corrected to diamonds.**

Line 426: Is this cited reference valid for the whole statement? If so, move to the end of the sentence. Otherwise it seems there are missing references relating to the highly-resistant endospores belonging to *Firmicutes* and *Actinobacteria*
**This reference was only intended for the opening statement regarding the physiochemical stresses faced by cells under frozen conditions. In the revision, we have added additional references to the end of the sentence citing the durability of Firmicutes endospores and Actinobacteria spores.**

Line 436-7: What data relates to this claim? Link to results if it relates to your study, otherwise cite relevant studies
**The citation was left out by mistake and has been added in the revised manucript.**

Line 592: Please swap "(46)" with a citation in a consistent style to the rest of the manuscript
**Corrected.**

**REVIEWER #2**

Doyle and Christner have produced new data documenting the taxa present and active in basal ice from two different glaciers in spatially distant locations. They supplement the data by integrating it into a meta-analysis of several other studies of microorganisms in ice. This is an important study, with regard to the new data alone, but the review/synthesis of existing data is something the biological glaciology community needs. The analysis is excellent despite the inconvenience of sequencing platform evolution during the 2010's when most the work was conducted. I enjoyed reading the paper and learned a lot from it while reading. I support publication of the manuscript. My comments and requests are focused on extracting more out of the data and for clarity of presentation. Alexander Michaud

General Comments:

Results: I would advocate the authors normalize their results in terms of a volume of ice. This allows for budgets to be made based on volumes of basal ice in a glacier. Normalizing results by grams of sample makes it more difficult to relate results back to the environment and to other studies. Is this frozen weight of ice currently? If you do want to keep it that form, is there a way to incorporate the density of the different basal ice types studied?

**Thank you for this comment. Normalizing by volume (mL) works well for debris-free glacial ice, but for ice that contains large sediment content, it artificially inflates biomass estimates because of their reduced water content. We contend that debris-rich ices are more analogous to a frozen soil environment like permafrost, which are typically normalized by mass (g). For this reason, we relate all data in the manuscript per gram of ice.**

Englacial ice: Do you mean meteoric ice throughout? This becomes important when you are classifying your samples for section 4.5. The glaciological community would likely argue that glacial ice begins at the bubble close off depth (see your line 560). Supraglacial and englacial may be better as "meteoric ice" or "glacial ice" and supraglacial should be distinguished between accumulation and ablation zone (see your line 547). I recognize this is a bit semantic, but some of the discussion dances around these ideas when trying to explain the results.

**Yes, supraglacial and englacial ice are both meteoric ice. However, we did not want to use a single category because microbes associated with meteoric ice exposed or near the surface have different sources than those in older meteoric ice with depth in a glacier. Relabeling all samples as meteoric and distinguishing those in the accumulation or ablation zone would not be sufficient to make this distinction. In addition, this suggested approach is difficult to implement for studies that do not report the zone samples were collected from.**

Specific Comments:

L29: When saying that basal ice is an important mechanism for dispersing microorganisms in the subglacial aquatic environment, I think the important point that needs to come along with this for the reader (especially the non-glaciologist) is that the freeze on or melting conditions vary spatially at ice sheet and glacier beds at multiple scales and thus represent a mechanism of dispersal.

**Agreed. We have revised the text accordingly.**

L37: When you say overlying englacial ice, do you mean meteoric ice? I think I understand that you are trying to say this is where the englacial microbes are, but stick to consistent terms (see general comment above).

**Please see our response above.**

L51: Laufer et al 2021 Nature Communications demonstrates this from Svalbard glaciers.

**Thank you. We have added this citation to the revised manuscript.**

Paragraph starting at L52: This paragraph oscillates between basal ice sediments and subglacial sediments which makes it confusing for the reader. With this said, the point of this paragraph is, I think, that knowing the subglacial conditions is important for understanding basal ice conditions because of the imprint subglacial conditions can have on basal ice. Without this connection, the focus on subglacial conditions leads to confusion. Please revise this paragraph.

**We agree and have revised this paragraph for clarity.**

L91: Please define temperate ice when used first in the introduction.

**Our use of "temperate conditions" earlier in the introduction on L27 was too vague. To address this, we have revised the text on L27 for clarity. As such, the**

**definition for a temperate glacier has not been changed since this is now the first use of "temperate ice" in the revised manuscript.**

L122: The last sentence of this paragraph is confusing. Please rewrite.

**Agreed. We have revised the sentence.**

L125: Were ATP samples melted at 4C like DNA extraction samples? Please clarify.

**Yes. We have added this detail to the text.**

L145: Describe your blanks.

**Description added to the text.**

L214: Perennially freshwater or "perennially frozen, freshwater environments"?

**Thank you. This typographical error has been corrected.**

L282: How were the potential contaminants flagged and removed? I missed this in the methods.

**This process is described at the end of section 2.6.**

Table 2: What is your limit of detection for the ATP method?

**The limit of detection for ATP using our approach was 10 pM, which represents ~0.01 pg ATP g$^{-1}$ ice. Measurements beneath this limit were thus listed as being <0.01 (pg g$^{-1}$) in Table 2.**

Figure 3: I am curious to hear your rationale for including the weathering crust of Matanuska Glacier into the englacial ice category. To me, a weathering crust is analogous to the active layer of permafrost environments, and active layer samples were excluded from this analysis. This is a seasonally thawed part of the glacier. Also, should it be englacial, seems supraglacial. Also, based on the Figure, then there would be two large clusters of supraglacial samples. I almost think the microbial communities are helping you to determine the sample cluster definitions rather than forcing a label on them a priori.

*"I am curious to hear your rationale for including the weathering crust of Matanuska Glacier…"*
**We agree the weathering crust is a very different system. Only samples from that study that had "glacial ice" as their listed isolation source in the NCBI metadata**

**were included. Please note that all samples originating from the "weathering crust aquifer" were excluded.**

*"I almost think the microbial communities are helping you to determine the sample cluster definitions rather than forcing a label on them a priori."*

**It does appear that in many cases, microbial community composition is indicative of ice type. However, we decided it would be more appropriate to label the ice samples *a priori* based on their physical descriptions rather than the composition of their microbiomes. Ice type certainly affects microbial community composition, while the vice versa is unlikely.**

- Also, I think the WISSARD samples need explanation that they were drill water sampled from the drill, which represents an integrated sample of firn and englacial ice given firn meltwater was used to start drilling.

**Agreed. We have added an asterisk to the WISSARD samples and noted this detail in the caption.**

- Some of these issues may be resolved by categorizing the samples as accumulation zone samples and ablation zone samples. Consider rerunning the cluster analysis with these categories.

**This is an interesting idea, but it would not meaningfully change anything within the analysis. The categories in Figure 3 are added afterwards for visualization; the clustering algorithm only "sees" sequence data and is agnostic to ice type. It would also be difficult to implement this new labeling scheme accurately as many studies do not report what zone of the glacier the samples were collected from.**

Figure 4: Can you clarify what the scale is for the x-axis of the right panels. Some taxa appear twice in the list. Is there something else to be learned here for taxa that appear more than once? Indicate in the legend that the lineage listed on the right list is the lowest and confident taxonomic assignment, which varies for some listed?

**The x-axis in the right panel represents the Euclidean distances of the scalar projections from the left panel. We have added this detail to the figure. Please note that following the reclassification with SILVA 138.1 (see our response to reviewer #1), deeper classification of several Cyanobacterial taxa in the meta-analysis resulted in there no longer being any duplicate lineages in this figure.**

L437: The sentence about pasteurization needs a citation.

**Citation added.**

L461: What is meant by "larger number of high-ratio ASVs"?

**We have revised the text so that high-ratio is explicitly defined as those ASVs having a 16S rRNA/rDNA abundance log-ratio >1.**

L533: By decline, you mean the dormant, strictly aerobic taxa are moving to the pool of dead microbes? Since you invoke that most all cells are already in a dormant state, then for a population to decline in this situation would mean die off, right?

**Correct. We have replaced "decline" with "die-off".**

L541-543: Given the shift to anaerobic taxa due to loss of oxygen, then there is a maximum $CO_2$ production that can occur in clean ice until dirty basal ice can provide alternative electron acceptors.

**Perhaps, but fermentation could be occurring, which is also a potential biotic $CO_2$ source in the ice.**

L572: Or maybe they were transported to the basal ice region from cryoconites above?

**This scenario is possible. However, since the marginal basal ice samples were exposed to the surface and sunlight when they were sampled, we feel our explanation is less speculative.**

FigureS2: Please insert a scale bar into the image.

**Adding an accurate scale bar is difficult because the imaged basal ice horizon is not aligned with the focal plane of the photograph. Instead, we have added a written description of the approximate scale of the sampled area to the figure legend.**

---

## Referee Report (RR1)

The manuscript is acceptable in its current form, I would like to clarify one of my comments and let the authors consider it.

My comment about volume was not a liquid volume, but rather a volume of basal ice (cm^3). I understand that g of basal ice is like a permafrost soil core and so volume (mL) doesn't make sense, but usually the permafrost soil also comes along with a soil water content and therefore a wet weight:dry weight ratio. I did not see any numbers on sediment content, which is important to know for biomass calculations. If you do want to stay with g of ice, that is fine, but is it g wet weight or g dry weight? The names of the different basal ice types are informative, but need to be comparable to the next study.

---

## Author Response (AR2)

Dear authors,

Thank you for your thorough response to reviews. As you know, one reviewer had a second look at the revised manuscript to provide some clarification for their previous comments. Their comment is below, and I therefore request a brief response and amendments to the manuscript if appropriate before we move to the final decision.

Thank you for your patience with the review process.

Dr Liz Bagshaw

Reviewer#1:

The manuscript is acceptable in its current form, I would like to clarify one of my comments and let the authors consider it.

My comment about volume was not a liquid volume, but rather a volume of basal ice ($cm^3$). I understand that g of basal ice is like a permafrost soil core and so volume (mL) doesn't make sense, but usually the permafrost soil also comes along with a soil water content and therefore a wet weight:dry weight ratio. I did not see any numbers on sediment content, which is important to know for biomass calculations. If you do want to stay with g of ice, that is fine, but is it g wet weight or g dry weight?

**We have added by-weight debris content to Table 2. We have also revised the caption to indicate that all measurements are normalized to sample wet weight.**

The names of the different basal ice types are informative, but need to be comparable to the next study.

**As described in the methods, the naming of the different basal ice types in our study follows the standardized classification scheme for basal ice defined in Hubbard et al. 2009.**